# Treatment of *Enterococcus faecalis* Infective Endocarditis: A Continuing Challenge

**DOI:** 10.3390/antibiotics12040704

**Published:** 2023-04-04

**Authors:** Laura Herrera-Hidalgo, Beatriz Fernández-Rubio, Rafael Luque-Márquez, Luis E. López-Cortés, Maria V. Gil-Navarro, Arístides de Alarcón

**Affiliations:** 1Unidad de Gestión Clínica de Farmacia, Instituto de Biomedicina de Sevilla (IBiS), Hospital Universitario Virgen del Rocío, 41013 Seville, Spain; 2Unidad Clínica de Enfermedades Infecciosas, Microbiología y Parasitología (UCEIMP) Grupo de Resistencias Bacterianas y Antimicrobianos (CIBERINFEC), Instituto de Biomedicina de Sevilla (IBiS), Hospital Universitario Virgen del Rocío/CSIC/Universidad de Sevilla, 41013 Seville, Spain; 3Unidad Clínica de Enfermedades Infecciosas y Microbiología, Grupo de Resistencias Bacterianas y Antimicrobianos (CIBERINFEC), Instituto de Biomedicina de Sevilla (IBiS), Hospital Universitario Virgen Macarena/SCIC/Universidad de Sevilla, 41009 Seville, Spain

**Keywords:** *Enterococcus faecalis*, endocarditis, antimicrobial treatment, resistances

## Abstract

Today, *Enterococcus faecalis* is one of the main causes of infective endocarditis in the world, generally affecting an elderly and fragile population, with a high mortality rate. Enterococci are partially resistant to many commonly used antimicrobial agents such as penicillin and ampicillin, as well as high-level resistance to most cephalosporins and sometimes carbapenems, because of low-affinity penicillin-binding proteins, that lead to an unacceptable number of therapeutic failures with monotherapy. For many years, the synergistic combination of penicillins and aminoglycosides has been the cornerstone of treatment, but the emergence of strains with high resistance to aminoglycosides led to the search for new alternatives, like dual beta-lactam therapy. The development of multi-drug resistant strains of *Enterococcus faecium* is a matter of considerable concern due to its probable spread to *E. faecalis* and have necessitated the search of new guidelines with the combination of daptomycin, fosfomycin or tigecycline. Some of them have scarce clinical experience and others are still under investigation and will be analyzed in this review. In addition, the need for prolonged treatment (6–8 weeks) to avoid relapses has forced to the consideration of other viable options as outpatient parenteral strategies, long-acting administrations with the new lipoglycopeptides (dalbavancin or oritavancin), and sequential oral treatments, which will also be discussed.

## 1. Introduction

Enterococcal infective endocarditis (IE) represents the third leading causal agent worldwide (10–15% of all cases) [1,2,3]. *Enterococcus faecalis* infective endocarditis represents 90% of enterococcal IE and has experienced important epidemiological changes in the last two decades. Patients with enterococcal IE are usually older, with higher rates of cancer, aortic valve affectation and previous history of urinary tract or abdominal infections [4,5]. In the past, enterococcal IE was mainly community-acquired, but nowadays there is a significant increase in the incidence of healthcare acquisition [6,7,8], particularly in aged patients (>70 years) with many comorbidities, one of them being colorectal neoplasms (advanced adenomas and carcinomas) [9,10]. Moreover, today *E. faecalis* is the most common causative organism isolated in IE in transcatheter aortic valve implantation (TAVI) [11]. 

Diagnosis of *E. faecalis* IE is challenging due to its often subacute course, with nonspecific constitutional symptoms and chronic anemia, difficult to interpret in an elderly and frequently immunosuppressed population with a large number of comorbidities. In fact, it is not uncommon that the first symptom that leads to diagnosis is left ventricular failure, being the rate of cardiac surgery reported around 40%, usually lower than clinically indicated due to the sometimes poor clinical condition of the patient [12,13,14]. 

Treatment of *E. faecalis* endocarditis has long been recognized as an important clinical challenge. The success of enterococcal population for surviving in multiple environments alongside a wide range of inhabitants, and the ease by which they acquire mobile genetic elements, including plasmids from other bacteria is surprising. Furthermore, the enterococci are frequently present within as bacterial biofilm (specially *E. faecalis*), which provides stability and protection to the bacterial population along with an opportunity for a variety of bacterial interactions and the acquisition of resistances [15,16]. The frequent lack of bactericidal activity of traditional agents (penicillin or ampicillin), the toxicity incurred with the addition of aminoglycosides and the increased reports of high-level resistances to them [17], in parallel with the production of bacterial biofilms over prosthetic devices [18,19], has led to a much higher rate of relapses (7–10%) compared with other etiologies [5,12,14,20]. These relapses can occur still several months after the end of the antimicrobial therapy [21,22], generating continuous uncertainty for the clinician and the need for a prolonged follow-up. Moreover, the emergence of multidrug-resistant isolates in *E. faecium* and the possibility of its transmission to *E. faecalis* brings a new concern for which we have yet no solid therapeutic evidence [23,24]. 

In this review, we will focus on the therapeutic options for *E. faecalis* IE in which we still have many therapeutic options, from “classical” guidelines to new alternatives. 

## 2. Mechanisms of Resistance

Enterococci exhibit significant resistance to a wide variety of antimicrobial agents that is relevant in most natural ecological settings in which enterococci inhabits. Enterococci are normal commensals of the human bowel and are routinely exposed to a plethora of antibiotics during many contemporary medical treatments. Their resistance plays a key role in the ecological dynamics that occur during and after any antibiotic therapy. Intrinsic resistance present in all members of the species differs from acquired resistance. The former is encoded within the core genome and the latter is present in only some members of the species and is obtained via the horizontal exchange of mobile genetic elements or by selection upon antibiotic exposure. Resistance for many antibiotics have emerged, including those that were traditionally useful to treat enterococcal infections, as well as those to which enterococci are incidentally exposed during therapy for infections caused by other bacteria.

A complete description of the mechanisms of resistance is beyond the scope of this review, and we will briefly review the most important ones for treatment. A complete description of the various types is provided in Table 1.

### 2.1. Beta-Lactams

Enterococci exhibit an intrinsic natural resistance to beta-lactams, due to the low affinity of their penicillin-binding proteins (PBP) for these antibiotics [25,26]. This intrinsic resistance differs among the different beta-lactams, with penicillins generally having the highest activity against enterococci, carbapenems having slightly lower activity and cephalosporins with the lowest activity, except new-generation cephalosporins such as ceftobiprole and ceftaroline [27,28,29]. Piperacillin activity is similar to penicillin, but oxacillin, ticarcillin, ertapenem or aztreonam have limited or no activity against enterococci. The most active penicillin in vitro is ampicillin, with a minimum inhibitory concentration (MIC) that ranges from 1 to 16 mg/L (much higher than most streptococci), usually one dilution lower than penicillin. However, despite an apparent good in vitro inhibitory activity (e.g., MIC = 1 mg/L for ampicillin), previous in vitro and in vivo studies promptly demonstrated that beta-lactam monotherapy was associated with a poor outcome in patients with endovascular infections [30,31]. Indeed, the bactericidal activity that is required for curation is rarely achieved with these compounds because certain “tolerance” (or lack of killing), that make the success of beta-lactam monotherapy unpredictable. Moreover, certain enterococcal strains are killed only at a specific concentration of the beta-lactam, above which the killing effect decreases (“paradoxical response”) [32], although the true significance in the real life of this in vitro effect is unknown [33].

While rare and not yet reported in Europe, resistance to beta-lactams in *E. faecalis* can be mediated by the production of a no-inducible beta-lactamase enzyme and may respond to a beta-lactamase inhibitor combination (e.g., ampicillin-sulbactam) plus an aminoglycoside, also maintaining sensitivity to carbapenems [34,35]. Non-beta-lactamase-mediated resistance to ampicillin is quite rare in *E. faecalis*. However, ampicillin plus beta-lactamase inhibitors and imipenem resistance has been reported and seems to be associated with mutations of the *pbp4* gene, that produce increased expression of low-affinity PBP4 or amino acid changes within the enzyme itself [36,37]. Conversely, resistance to beta-lactams in most clinical isolates of *E. faecium* is associated with mutations or overproduction of PBP5, with ampicillin MICs > 256 mg/L in some strains [38]. 

### 2.2. Aminoglycosides

Enterococci, due to its outer bacterial wall, is relatively impenetrable to aminoglycosides and are then considered “structurally resistant” to clinically achievable concentrations of these antibiotics [39]. Most species show low-level aminoglycoside resistance (gentamicin MIC < 1024 mg/L and streptomycin MIC < 512 mg/L). However, the combination with cell wall-active agents that blocks peptidoglycan synthesis raise the permeability of the enterococcal wall and markedly increases the uptake of these antibiotics, thus promoting synergy between beta-lactams or vancomycin with gentamycin or streptomycin, with good clinical results [40,41,42]. 

However, the existence of high-level resistance (HLAR) does not allow the use of this combination. This acquired resistance include alterations of the aminoglycoside’s ribosomal target due to chromosomal mutation (streptomycin) [43] and plasmid-mediated resistance genes that encode various aminoglycoside-modifying enzymes, which results in the development of a very high resistance (MICs usually > 2000 mg/L) [44].

The inactivating enzymes may be phosphotransferases, acetyltransferases or nucleotidyltransferases. The most commonly found enzyme is the bifunctional enzyme AAC(6′)-Ie-APH (2″) that confers resistance to all available aminoglycosides, excepting streptomycin. Other enzymes frequently found in HLAR enterococci are ANT (6′)-Ia and APH (2″)-Ic which are the responsible of resistance to streptomycin and gentamycin respectively. In general, resistance to streptomycin is restricted only to this drug, while resistance to gentamycin implies resistance to all other aminoglycosides except for streptomycin.

### 2.3. Glycopeptides

Enterococci are considered susceptible to vancomycin and to teicoplanin, that have a long elimination half-life which permits once-daily dosing and has the advantage of much lower renal toxicity.

Strains of enterococci are considered sensitive to vancomycin (MIC < 4 mg/L), intermediate (MIC = 8–16 mg/L) or fully resistant (MIC > 16 mg/L). The resistance is due to the acquisitions of operons that alter the nature of peptidoglycan precursors, substituting a D-lactate of the terminal D-alanine in the UDP-MurNac pentapeptide. Glycopeptides bind to the terminal D-alanine of the cell wall precursor, preventing PBP access, but pentapeptide stems terminating in D-lactate have a 1000-fold lower affinity for vancomycin. Different genotypes with resistance to vancomycin and teicoplanin have been described, being the operon *van*A the most encountered in the clinical setting (Table 1).

The isolation of vancomycin-resistant enterococci (VRE) has increased since 1986 in all the world and nowadays is prevalent (60–80%) among *E. faecium* isolates in USA [45]. In Europe, VRE isolates are common in farm animals, feed, and wastewater, and as colonizers in healthy humans [46], but are much less frequent in hospitalized patients (although with high variability between countries), probably due to the widespread use in the livestock industry of the glycopeptide avoparcin, as a growth promoter [47]. However, even after the prohibition of avoparcin use, the European continent has continued with an important increase in the isolation of VRE (mostly *E. faecium*) from hospitals, probably due to other factors that are promoting the dissemination of VRE isolates, such as hospital clonal clusters, like CC17 [48].

In the case of *E*. *faecium* and some strains of *E*. *faecalis*, *vanA* and *vanB* genes play a major role. Fortunately, *E*. *faecalis* vancomycin-resistant are usually susceptible to beta-lactams, as are *E*. *gallinarum* and *E*. *casseliflavus* (which are intrinsically vancomycin-resistant).

### 2.4. Daptomycin

Daptomycin is a lipopeptide antibiotic approved for the treatment of complicated skin and soft tissue infections and *S. aureus* bacteremia in adult patients, including those with right-sided infective endocarditis. The mechanism of action involves the interaction of the antibiotic with the cytoplasmic membrane via the calcium-dependent insertion, leading to a variety of alterations in cell membrane characteristics. Daptomycin has dose-dependent bactericidal activity against most Gram-positive agents, including vancomycin and ampicillin-resistant enterococci [49]. The Clinical and Laboratory Standards Institute (CLSI), has recently determined a new “susceptible” breakpoint of ≤2 mg/L for *E*. *faecalis* and a separate “susceptible dose-dependent” breakpoint of ≤4 mg/L for *E*. *faecium*, but indications do not include VRE [50]. However, EUCAST (European Committee on Antimicrobial Susceptibility Testing) daptomycin breakpoint have not been set due to various uncertainties, particularly the inability of, even the highest published doses (12 mg/kg/day), to achieve adequate exposure against all wild-type isolates of *E. faecalis* and *E. faecium* [51]. In fact, emergence of daptomycin-resistant strains with treatment failures has been described with standard dose monotherapy (6 mg/Kg) [52,53]. Resistance to daptomycin occurs through a variety of mutations that have different effects depending on the species. Much of it is attributed to mutations in several genes including the stress-sensing response system YycFGHIJ and LiaFSR, an also alterations in phospholipid biosynthesis enzymes such a cardiolipin synthetase *cls* and glycerophosphoryl diester phosphodiesterase *gdp*D [54,55].

### 2.5. Quinolones

The activity of fluoroquinolones against enterococci is moderate and resistance is frequent among clinical isolates. Ciprofloxacin and levofloxacin have marginal activity against enterococci and moxifloxacin is more potent against Gram-positive bacteria but exhibits only intermediate activity versus enterococci [56]. Mutations in the *parc* gene encoding the parC subunit of topoisomerase IV are the first step in the acquisition of resistance, which may be followed by additional mutations in the *gyr*A gene encoding the GyrA subunit of DNA gyrase, thereby increasing the level of resistance [57]. In general, most resistant strains have mutations in the two genes that are related to aminoacidic changes in the Ser83 position of DNA gyrase and in the Ser80 position of topoisomerase IV. Low-level resistance may also be due to alterations in the uptake of these antimicrobials into the bacteria, although specific efflux pumps have not been identified [58].

### 2.6. Oxazolidinones

Linezolid is the most common used agent of this class. This drug selectively binds to the 50S ribosomal subunit, resulting in inhibition of bacterial protein synthesis. In general, this resistance is still rare (overall 1–2%) but has been described in *E. faecium* and especially in *E. faecalis,* with higher prevalence in USA and Africa [59]. The resistance is due to mutations in the 23S subunit of ribosomal RNA and ribosomal protein-coding regulatory genes such as *rpl*C, *rpl*D, and *rpl*V, in which mutations lead to amino acid substitutions on several ribosomal proteins. Enterococci possess multiple copies of the gene encoding this subunit and the higher the number of mutated alleles in this gene, the higher the level of resistance: with a single mutated gene, the MIC of linezolid is 4–8 mg/L, while with five mutated alleles, the MIC rises to 64 mg/L [60]. Moreover, enterococci strains have also exhibited the acquisition (via plasmid) of more generic resistance genes such as *cfr* or *cfr*(B), which encodes a chromosomal methylase that modifies bacterial 23S rRNA [61]. This enzyme confers resistance to various antimicrobial classes, including phenicols, lincosamides, oxazolidinones and streptogramin A, and a decreased susceptibility to spiramycin and josamycin macrolides. Finally, plasmid-mediated resistance has also been attributed to the acquisition of *optr*A, which encodes a putative ABC (ATP-binding cassette) transporter [62]. Most of the reported cases are from patients who had received linezolid for long periods and were selected in the presence of the antibiotic, although clonal dissemination has also been described [63].

### 2.7. Tigecycline

This bacteriostatic drug inhibits protein synthesis through an interaction with the bacterial 30S ribosomal unit, blocking bacterial protein synthesis. That confers a broad-spectrum therapeutic effect against multi-drug-resistant Gram-positive bacteria including VRE and MRSA, in addition to beta-lactamase-producing bacteria. Very few resistances have been reported for *E. faecalis* and *E. faecium*, although the emergence of resistant strains seems being increasing in Europe and Asia [59]. Mutations in various efflux pumps is the main mechanism, although other resistance-related mechanisms are deletions in ribosomal protein gene *rpsJ* and elimination of transcriptional regulation of the ribosomal protection protein [64].

## 3. Therapeutic Choices

The management of enterococcal IE has long been recognized as a challenging clinical problem. Endovascular infections, such as IE are entities in which bactericidal therapy are of paramount importance for eradication of infecting organisms before further damage occurs (embolism production, valvular destruction). However, unlike the clinical success initially observed with penicillin in the treatment of staphylococcal and streptococcal IE, failure rates with this antimicrobial in enterococcal IE was unacceptable (up to 20%) [30,31]. As it was referred above, the poor performance of penicillin monotherapy has been attributed to the “natural tolerance” of many enterococcal isolates to beta-lactams, which means that they do not achieve a bactericidal effect, even though they inhibit enterococcal growth and are successful in other infections, such as catheter-related bacteremia and those from urinary tract [25,26].

The actual recommendations stated in international guidelines [65,66] are provided in Table 2. We will focus on them and will consider new alternatives that are described in Table 3. Renal adjustments for antimicrobials are provided in Table 4.

### 3.1. Beta with-Lactams with Aminoglycosides (A + G)

The combination of penicillin plus streptomycin was empirically found to cure the patients who were not improving with penicillin alone and was subsequently shown to have synergistic bactericidal activity in vitro. The development of high resistance to streptomycin (which abolishes synergism) led to the use of gentamycin, an aminoglycoside for which resistance was rare at the time and showed similar results in terms of bactericidal effect and clinical efficacy. Treatment of enterococcal IE with the combination of penicillin plus streptomycin or gentamycin has been evaluated in many studies and became the standard of care many decades ago for patients with IE due to enterococci in the absence of HLAR strains [30,40,41,42,99,100].

### 3.2. Dual Beta-Lactam Therapy (A + C)

Another option that is especially recommended in elderly people, patients with previous renal impairment, and specially, in IE caused by HLAR strains, is the ampicillin + ceftriaxone (A + C) regimen, which is synergistic *in vitro* and has also proven effective both in experimental studies and in real life. Its similar efficacy and lower toxicity have led to be included as an alternative regimen in the current guidelines [65,66].

In 1995, a French group described an unexpected in vitro synergy between amoxicillin and cefotaxime, an antimicrobial discovered in 1981 [101]. Enterococci are naturally resistant to cephalosporins, which act only on Peptide Binding Proteins (PBP) 2 and 3, triggering the production of more efficient PBPs 1, 4 and 5 under treatment. However, aminopenicillins (ampicillin and amoxicillin) and ureidopenicillins (piperacillin) act effectively on these other PBPs, which means that the joint use of both classes of drugs results in a complete blockage and synergy of action in inhibiting bacterial growth. These results were considered by a Spanish group that found identical synergy between ampicillin and another cephalosporin (ceftriaxone) discovered years later with a very favorable pharmacokinetic profile that allowed less frequent administration, due to its high plasma half-life. Gavaldá et al. in 1999 demonstrated a reduction of 1 to 4 dilutions in the MIC of ampicillin of 10 strains of HLRA *E. faecalis* when using the fixed dose of 4 mg/L of ceftriaxone by the double-disk technique and of at least one dilution when using micro dilution in Mueller-Hinton medium for its determination [102]. Using time-death curves, an ampicillin concentration of 2 mg/L and varying concentrations of 5–60 mg of ceftriaxone, they achieved a >2 log reduction of the initial inoculum at 24 h of incubation in all strains, and this effect increased to >3 log (synergy) in 7 of the 10 strains when 10 mg/L doses of ceftriaxone were used, and in six strains when the concentration was 5 mg/L. Similar results were reported much later, showing that even ampicillin concentrations at 1 mg/L + 2 mg/L ceftriaxone were synergistic [103]. Using a humanized model in the experimental animal, the authors found that by administering the equivalent of 2 g of IV ceftriaxone, drug concentrations at 12 h were around 50 mg/L and 20 mg/L at 24 h (antibiotic bound to proteins). As ceftriaxone bound to protein at 90%, the administration of 2 g IV/12 h. of ceftriaxone, together with the administration of ampicillin, could always achieve free drug concentrations of ceftriaxone above 2–4 mg/L, guaranteeing this synergistic effect during the entire dosing interval. These facts were effectively translated into a clear decrease in the colony count in the vegetations of the experimental animals treated with this regimen, and even the complete sterilization of the vegetations in some animals infected with certain strains.

The translation of this elegant experimentation on the animal model was published eight years later by the same group in a multicenter trial in 13 Spanish hospitals, in which 21 patients with HLRA strains and 22 with non-HLRA strains at high risk of nephrotoxicity were treated [104]. Cure was obtained in 100% of patients with HLRA strains who completed the protocol with this regimen. Due to the small sample size and the failures obtained in non-HLRA strains, the guideline was promptly considered but only for HLRA strains, due to the lack of existing alternatives. However, in 2003 the same group demonstrated, again in the experimental animal model, the efficacy of this pattern also in non-HLRA strains [105]. Again ten years later, the demonstration of its clinical efficacy in these non-HLRA strains compared to the “standard” treatment (ampicillin + gentamicin) came from a Spanish multicenter trial, in which 150 patients treated with the A + C regimen (ampicillin + ceftriaxone) were compared to 87 patients treated with A + G (ampicillin + gentamicin), showing equal mortality (22% vs. 21% during hospitalization and 8% vs. 7% at 3 months) and recurrences (3% vs. 4%), but with lower nephrotoxicity (23% vs. 0%; *p* < 0.01) [106]. Following these excellent results, the American and European guidelines included this regimen as “preferred” in HLRA strains and “alternative” in non-HLRA strains [65,66]. However, in Europe and especially in Spain, the A + C regimen progressively gained followers until it became the majority [107], and this was easily explained by its lower toxicity. The profile of the patient with enterococcal IE is often an elderly person, with abundant comorbidities and in many cases previously weakened by a previous hospitalization, and it is precisely in this group where the physician must be especially cautious about the side effects of treatment [4,5,6,13,14].

One concern with ceftriaxone use is that is has been pointed as an independent risk factor for *Clostridioides* infection [108] although no such complication has been reported from Spain in which this guideline is prevalent. Another concern is that some clinical and observational studies implicate the use of ceftriaxone as major risk factor for occurrence of vancomycin-resistant *E. faecium* infection, including bacteremia [109,110]. In animal studies, ceftriaxone use promotes gastrointestinal colonization by VRE, probably due to the high biliary excretion of ceftriaxone that could select for drug-resistant enterococci living there [111,112]. Unlike ceftriaxone, other cephalosporins antibiotics, such as cefepime [111], and ceftaroline [113] do not appear to promote VRE colonization, and the combination of ampicillin plus ceftaroline have demonstrated efficacy similar to ampicillin plus ceftriaxone in several pharmacodynamic studies, although no clinical data are yet available [69,70]. Ceftobiprole have high affinity for enterococcal PBPs and have demonstrated efficacy against VanB-resistant *E. faecalis* in addition to synergy when used in combination with aminoglycosides, but this combination requires further exploration in human subjects [28].

### 3.3. Glycopeptides

Vancomycin is an alternative therapy recommended in European and American guidelines for patients unable to tolerate penicillin or ampicillin. Vancomycin reduced CFU/mL in vegetations significantly more than ampicillin monotherapy in the rabbit experimental model [82]. However, combinations of penicillin or ampicillin with gentamicin are preferable to combined vancomycin-gentamicin because of the potential increased risk of ototoxicity and nephrotoxicity with the vancomycin-gentamicin combination during six weeks. Moreover, combinations of penicillin or ampicillin and gentamicin are more active than combinations of vancomycin and gentamicin in vitro and in experimental models of IE [114].

Teicoplanin is particularly interesting due the in vitro data that demonstrate advantage over vancomycin against *E. faecalis*, with MIC_90_ values usually lower than that for vancomycin [115,116,117]. Furthermore, its long-elimination half-life permits once-daily dosing and exhibit concentration-dependent activity with excellent results in experimental studies combined with gentamicin [118,119] and much lower toxicity than vancomycin [71,72,120]. Several observational studies (overall 56 patients) support the use of monotherapy with teicoplanin at doses of 6–10 mg/kg/d (two of them also introduced a loading dose), mainly as a continuation treatment for *E. faecalis* IE when adverse events have developed with standard treatments, or to facilitate outpatient treatments [73,74,75]. The largest study conducted supported the use of monotherapy with teicoplanin for treating *E. faecalis* IE as continuation therapy. The reported mortality related to IE was low (8%), although the population treated with teicoplanin suffered from less severe IE than the standard therapy group [74]. Within the patients treated with teicoplanin as a continuation or salvage therapy, 16 died (32%) in a minimal follow-up period of 3 months. Only three relapses were reported in these studies. Then, favorable results and very few toxicities lead us to consider it as a reasonable alternative. Theoretically, teicoplanin has also activity against enterococci with *Van*B mediated resistance, but development of resistance during therapy is concerning [121] and cannot be a recommendation in this setting.

### 3.4. Daptomycin

Although there are no prospective randomized-controlled studies evaluating the efficacy of daptomycin for the treatment of E. faecalis IE, several reports including a total of 26 patients were published shortly after its approval, within an “off-label” use [122,123,124]. The treatment scheme was considerably heterogeneous, included initial and salvage therapy, monotherapy and combined regimens, and the mean doses ranged between 8.5 and 10.125 mg/kg/day. Mortality rates reported were low (0–22%), although only one study [123] attained more than a one-month follow-up. In one study, the salvage treatment of five *E. faecalis* IE episodes was reported [122], of which four needed a treatment change due to treatment failure. Daptomycin patients had longer duration of bacteremia (6 versus 1 day) and greater need of therapy switch due to complications (66.7% versus 0%) compared with conventional antibiotic regimens (ampicillin or vancomycin ± gentamicin), although there were no differences regarding duration of hospital stay or mortality. So, the stated final conclusions differed, with two supporting daptomycin as an alternative treatment in this scenario [123,124], and one showing some concerns [122]. Among 22 patients with enterococcal IE treated with daptomycin in a European registry (18 *E. faecalis*), the success rate was 73%, but no information regarding dosage or combination therapy was given [125]. An observational prospective single center study found similar outcomes in patients with enterococcal endocarditis treated with daptomycin-based regimen versus standard regimens, although daptomycin was used in combination with another antibiotic (mostly a beta-lactam) an at high doses (>10 mg/kg/day) [124]. Microbiological failures of daptomycin were promptly reported when “standard” doses (6 mg/Kg/day, approved dose for *S. aureus* bacteremia and right-sided IE) were administered, and high doses (8–12 mg/Kg/day) are now recommended for enterococcal and *S. aureus* severe infections [50,51,52,53]. It is important to note that the daptomycin MIC90 for enterococci is higher than that of staphylococci (4 mg/L and 0.5 mg/L, respectively), supporting the concept that higher doses of daptomycin may be needed for the management of enterococcal IE, and in vitro studies have demonstrated that a high percentage (33%) of *E. faecalis* incubated with daptomycin at a subinhibitory concentration (2 mg/L) can develop MIC ≥ 8 mg/L [126]. Daptomycin display a dose-dependent bactericidal effect and high-dose regimens have demonstrated an enhanced pharmacodynamic profile, and perhaps the most bactericidal regimen against VRE [127]. However, microbiological failures also have been described with high doses in patients with prior daptomycin exposures, prostheses, or immunocompromised patients with long hospitalization courses [53,128,129]. Therefore, this alternative could be considered in resistant isolates or when adverse events appear, but not to simplify antibiotic treatment. Taking also account the synergistic activity between daptomycin and beta-lactams [83,84,130] fosfomycin [85,86,87,88,89,90,91] or tigecycline, a combination regimen with high doses seems to be preferable, whereas single therapy with this drug should be used with caution.

### 3.5. Fosfomycin

In vitro data have demonstrated synergy with fosfomycin in combination with ceftriaxone, rifampin, tigecycline, daptomycin and teicoplanin [85,86,87,88]. Current oral fosfomycin use is limited to uncomplicated urinary tract infections due to limited absorption and intravenous formulation are yet unavailable in the USA. However, fosfomycin has demonstrated utility against MSSA and MRSA endocarditis in combination with daptomycin or imipenem [89,90,91,131] and a study of in vitro and in vivo with the guinea pig model using intraperitoneal fosfomycin demonstrated promising activity against both planktonic and biofilm-forming *E. faecalis* when the drug was used in combination with gentamicin and daptomycin [132]. Thus, new therapeutic options with this drug could be considered in the future for *E. faecalis* IE.

### 3.6. Linezolid and Tedizolid

Linezolid after a few promising studies has been recommended for the treatment of endocarditis as result of multi-drug resistant enterococci [133] and has been recommended in the USA for the treatment of Enterococcus species caused by strains resistant to penicillin, aminoglycosides, and vancomycin [65]. Regrettably, widespread use from the year 2000 has result in an emerging of linezolid-resistant VRE in 2001 [134] and increasing of these strains especially in hospitals from various countries (Denmark, Spain, Germany…) [135]. However, the use of linezolid is a matter of controversy because of the lack of bactericidal effect and the lack of randomized clinical trials or robust cohorts. Linezolid has displayed efficacy in the treatment of VR *E. faecium* bacteremia with an open-label nonrandomized, compassionate-use program reporting microbiological and clinical cure rates of 85.3% and 79% respectively with 10 out of 13 patients with VRE strains (76.9%) achieving clinical cure in the subgroup of endocarditis [136]. A systematic review attempted to evaluate the clinical efficacy of linezolid in the treatment of enterococcal IE. This study found that 7 out of 8 cases improved or were cured with linezolid: four of the included cases were caused by *E. faecalis* (two VRE) and the rest of them were cases of IE vancomycin-resistant *E. faecium* [137]. But clinical evidence is supported only by these case reports with heterogeneous results and numerous cases of enterococcal infections resistant to linezolid have been reported [138,139,140]. In a Danish cohort of consecutive IE patients 38 out of 550 patients were treated with Linezolid [141]. The authors retrospectively compared individuals who had received this antimicrobial with a control group, and no significant differences regarding in-hospital mortality or at one year of follow-up were detected. Some authors reported similar results in small single hospital cohorts [142,143] although they did not specify the time of initiation and duration of the linezolid therapy nor its effect on IE. Recently, an interesting study reported from the Spanish cohort (GAMES) retrospectively analyzed 295 consecutive IE treated with linezolid, 38 of them enterococci IE [144]. In this cohort, in-hospital mortality in patients treated with linezolid was higher than in controls, and as determined with the multivariate analysis, linezolid was an independent risk of mortality. One of the principal drawbacks is the need of a prolonged treatment (six weeks) of *E. faecalis* IE that usually occurs in elderly population, which is more likely to the myelotoxicity and neuropathy produced by this drug.

Gastrointestinal disorders and myelotoxicity are less frequent with tedizolid, and a favorable action against MRSA IE have been reported in experimental models (rats and rabbits). Against VRE tedizolid has a fourfold lower MIC when compared to linezolid and has activity against linezolid-resistant strains with a *cfr* mutation, probably due to additional interactions with the ribosomal [95,96,145,146,147]. Thus, compared to linezolid, tedizolid has the potential to be a first-line agent for the treatment of serious VRE infections, but until now, no clinical data have been published in patients with IE.

### 3.7. Quinolones

Fluoroquinolones have been used in the treatment of some enterococcal infections such as chronic enterococcal prostatitis with relapsing bacteremia. Like the tetracyclines, these antibiotics have also been used as part of combination therapies in endocarditis. The combination of ampicillin plus ciprofloxacin was tested in an experimental model of rabbit endocarditis with *E. faecalis* and the regimen caused a significant decrease in bacterial counts compared to each compound alone, although it was less effective than the combination of beta-lactams and aminoglycosides [67], and this effect was previously reported in vitro [68]. Additionally, the use of ampicillin plus ofloxacin was shown to be also synergistic in vitro, achieving bactericidal activity, and to successfully clear the bacteremia in a patient with *E. faecalis* IE exhibiting HLAR [148]. Nonetheless, the lack of clinical experience and the increased rates of resistance to some of these compounds usually preclude the use of these antibiotics for *E. faecalis* IE, particularly as monotherapy.

Delafloxacin is a new quinolone active in vitro against MSSA, MRSA, CoNS and streptococci and interestingly retains activity against fluoroquinolone-resistant strains. Specific features in the delafloxacin molecule determines enhanced activity in acidic environment due to its anionic character, which eventually leads to improved activity [149]. Delafloxacin has been recently approved for acute bacterial skin and skin structure infections [150] and for the treatment of community-acquired bacterial pneumonia [151]. Delafloxacin can be given intravenously or orally due to its good bioavailability (60–70%) [149]. However, according to EUCAST, there is insufficient evidence that enterococci are a good target for therapy with delafloxacin and no clinical data have been published on the use of delafloxacin for IE.

### 3.8. Tigecycline

Tigecycline is a broad-spectrum antibiotic derived from minocycline which is approved for skin and soft tissue infections, including those with vancomycin-susceptible *E. faecalis*. In the treatment of soft tissue infections (including those with vancomycin-susceptible *E. faecalis*), tigecycline sowed a microbiological eradication rate of 87.5%, similar to vancomycin plus aztreonam [152]. In complicated abdominal infections, tigecycline exhibit similar rates of microbiological curation for vancomycin-susceptible *E. faecalis* (78.8%) than imipenem [153]. Moreover, some in vitro models suggest that synergism of tigecycline with vancomycin, gentamicin and rifampin can be achieved for certain strains of *E.* faecalis and *E. faecium* [154] and successful therapy of endocarditis was reported with the combination of tigecycline plus daptomycin in several cases [92,93,94]. However, a serious drawback of the use of tigecycline monotherapy is the low levels obtained with this antibiotic [155] and the emergence of resistance during therapy has been reported in experimental studies [156]. Thus, the use of this compound as monotherapy for severe enterococcal infection is discouraged, although its role in combination therapies with bactericidal effects warrants further investigation.

### 3.9. Dalbavancin and Oritavancin

Considered a subclass of the glycopeptide antibiotics, the new lipoglycopeptides have similar mechanisms of action of binding to the carboxyl terminal d-alanyl-d-alanine residue of the growing peptide chains but differ from their parent glycopeptides by the addition of lipophilic tails that allows prolonged half-lives, giving them unique dosing profiles.

Dalbavancin has a long-acting parenteral administration due to its high-protein binding (93%) and prolonged elimination half-life (14.4 days), that allows prolonged intervals between doses (7–14 days) [157] A promising activity against Gram-positive biofilms has also been reported [158].

Dalbavancin was approved in the USA and Europe to treat acute bacterial skin and skin structure infections caused by Gram-positive cocci isolates, including vancomycin-susceptible *E. faecalis*, but it must be remarked that it is inactive against *Van*A phenotypes. Although it has not been approved to treat patients with bloodstream infections or IE, there are in vitro studies showing a good susceptibility (MIC90: 0.06 mg/L) of most E. faecalis isolates (97.6%) collected from patients with a diagnosis of bacterial endocarditis [159]. Off-label utilization of dalbavancin was extensively done in patients with osteomyelitis, joint infections, and articular prostheses, and less in cardiovascular infections [76]. A retrospective cohort in Austria evaluated 27 adults with Gram-positive bacteremia with IE treated at least with one dose of dalbavancin with excellent results [77]: In most patients dalbavancin was used as sequential treatment after clearance of bacteremia to allow a promptly discharge with outpatient treatment, and the same scheme was used in a Spanish cohort that included 34 patients (three of them with *E. faecalis* IE) [78]. Two dosing strategies are used with similar results: a 1000 mg loading dose and the 500 mg/week or a 1500 mg loading dose and then 1000 mg every two weeks and in these cohorts, six patients were successfully treated. Thus, limited available evidence suggests that dalbavancin might be a good option to treat *E. faecalis* IE in the context of sequential outpatient therapy, but studies with full use (not only sequentially) of this drug throughout treatment are needed.

Oritavancin is an interesting drug, very active against Gram-positive cocci including enterococci, and that also retains some activity against *Van*A and *Van*B-mediated vancomycin resistance. Among two collections of more than 10,000 isolates, oritavancin showed potent in vitro activity against staphylococci (including MRSA), streptococci and enterococci [160,161]. Although higher MIC were registered against vancomicn-resistant E. *faecalis* (vanA phenotype) than against vancomycin-susceptible strains, all VanA- resistant *E. faecalis* were inhibited at 0.5 mg/L or less. Its high protein-binding (85–905) and the prolonged terminal half-life (200–300 h) permits the administration of a single dose of 1200 mg with good therapeutical levels beyond two weeks [162] and a good activity in biofilms [163]. After the initial dose of 1200 mg, sequentially doses of 800 mg can be administered weekly for infections that will require a more prolonged treatment such as osteomyelitis [164].

Elimination of oritavancin mainly occurs through the reticuloendothelial system and no adjustments of dosage are needed in the cases of renal or hepatic failure. Oritavancin is a weak inhibitor of CYP2C9 and CYP2C19, and an inducer of CYP3A4 and CYP2D6, thus drug-drug interactions (e.g., patients treated with warfarin) should always considered. Attention should be paid to the possible alterations of some coagulation tests after oritavancin administration because of its interaction with the phospholipid reagent. Prolonged prothrombin and active partial thromboplastin times have been reported and thus, the use of intravenous unfractionated heparin sodium is contraindicated for up to 5 days after oritavancin administration owing the inability to reliably monitor coagulation tests. However, the results of chromogenic factor Xa and the thrombin time assays are not affected, allowing the use of fractionated heparin.

Oritavancin was also approved in the USA and Europe to treat adults with acute bacterial skin and skin structure infections caused by Gram-positive cocci isolates, including vancomycin-susceptible *E. faecalis*, but several reports with its utilization in osteoarticular infections, bacteremia and very few endocarditis have been reported [79,80,81,165].

In conclusion oritavancin seems also an important option for outpatient therapy and early discharge in patients with *E. faecalis* IE, but very limited number of papers are available, especially in this setting, and there is no experience with quantity and dosing schedule and duration of treatment. Thus, further studies are necessary for optimizing and refining its place in the treatment of IE.

## 4. Duration of Treatment

One of the difficulties in the treatment of *E. faecalis* IE lies in its prolonged duration (4–6 weeks) and need to combine two antimicrobials to achieve synergy, which must also be administered in several doses/day. Treatments of less than six weeks have been associated with a greater number of recurrences [166], although in certain patients (endocarditis on native valve and absence of paravalvular extension) four weeks are probably enough [167]. A shorter treatment with aminoglycosides has already been mentioned by some authors [168] and was later confirmed by a Swedish study [169] in which no differences were found between 2 and 4 weeks of administration. This same fact was endorsed by a Danish study [170] which, when administered over a short period (2 weeks), also found no difference between intermittent administration and administration in a single daily dose. No differences between 2 or 4 weeks were found in complications, relapses, in-hospital mortality, or 1-year event free survival, but patients receiving 2 weeks treatment with gentamycin therapy did not experience a significantly reduction in renal function at discharge, compared to those receiving the full course, as measured by estimated glomerular filtration. Although this study was limited by a small size and insufficient power to establish the optimal duration of aminoglycoside treatment, other studies have shown that gentamycin nephrotoxicity increases with duration of treatment [171] and its discontinuation occurred frequently after 14–18 days of treatment [106,107]. It is also be noted that the decrease in renal function is only partly reversible [172,173]. Then, a 2-week treatment course of aminoglycosides seems reasonable taking account the clinical picture of most patients with *E. faecalis* IE (elderly, fragile and with high degree of comorbidity) [5,12,14,174].

## 5. Outpatient Parenteral Antimicrobial Therapy (OPAT)

Outpatient parenteral antimicrobial treatment constitutes one of the most recent advances in the treatment of infective endocarditis. After the first two weeks, the septic process is usually well controlled (lack of fever and negative blood cultures), the risk of embolism is dramatically reduced [175,176], and the possible complications derived from the structural damage of the heart have been properly assessed by echocardiographic studies. Then the possibility of further complications is greatly reduced, and home treatment brings undeniable advantages in terms of comfort for the patient and their environment, cost savings as well as avoiding nosocomial infections [177]. In 2019, the GAMES group (Grupos de Apoyo al Manejo de la Endocarditis en España) established much less strict criteria than the original recommendations for this therapeutic approach [178,179], but which proved to be equally safe and allow outpatient treatment of *S. aureus* (including methicillin-resistant strains, MRSA) and *Enterococcus* spp [180].

The “standard” guidelines (A + G) recommended in both American and European guidelines make this option almost impossible on an outpatient basis, since they recommend the administration of aminoglycosides every 8–12 h and monitoring the levels. We would therefore need two routes (one of which would be for the continuous or intermittent administration of ampicillin by means of a perfusion pump) or a well-trained family member capable of performing (two or three times a day) these manipulations, in addition to the logistics inherent to the monitoring of levels. However, these recommendations are based on experimental studies [181,182,183,184], sometimes contradictory [185,186] possibly due to differences related to the different pharmacokinetics of the experimental model and humans. On the other hand, there are abundant studies that show the efficacy of single-dose administration of aminoglycosides, with excellent activity due to their prolonged post-antibiotic effect and much less renal toxicity [187]. Therefore, the translation of these experimental results to humans, as reflected in the guidelines, is far from being justified by a hypothetical greater efficacy, considering the known increase in toxicity of aminoglycosides in intermittent administration for 4–6 weeks [188]. Therefore, a valid option would be the administration (continuous or intermittent every 4 h) of ampicillin by pump, in addition to a single dose of gentamicin, which could be done at the time of refilling the pump.

The A + C regimen arises from the finding of synergy between the two drugs, when free drug levels of ceftriaxone between 5–10 mg/L are achieved. Based on experimental data derived from the humanized rabbit model, doses of 2 g/12 h in humans are recommended [105]. For OPAT programs, this implies the difficulty of an administration of ceftriaxone every 12 h, which is not always easy due to the availability of the patient or caregiver to manipulate the system. Our group, based on theoretical concentrations of 30 mg/L (total drug) at 24 h when a single dose of 4 g was used, began to administer this regimen (A + C24) in enterococcal IE, with good initial results [189]. In parallel and given the absence of pharmacokinetic data with this dose, we conducted a clinical trial with healthy volunteers where we analyzed the pharmacokinetics of ceftriaxone administration at a dose of 2 g. IV/12 h (C12) and 4 g. IV/24 h (C24). Unfortunately, we observed that administration of a single dose of 4 g. did not achieve the target levels above 5 mg/L of free drug in any subject at 24 h, in very few at 18 h, and only in half at 12 h [190]. As previously reported in trials in healthy volunteers, administration every 12 h did not achieve these levels at 24 h in most individuals, although it did at 18 h in half of them [191,192]. However, the good clinical results reported with this regimen (A + C12) suggested that these “target” concentrations of ceftriaxone might not be necessary during the whole time but at least during 75% of the dosing interval, which could be achieved only with the administration every 12 h. Administration every 24 showed clearly poor pharmacokinetics, and further dose escalation (6–7 g) did not seem to be a good solution either, as we tested in Monte Carlo simulation models. The binding of ceftriaxone to plasma albumin is very high (90%) and dose-dependent, but saturable, and administering a very high dose of the drug to over 100 mcg/mL would result in high levels of free drug after infusion. This effect could be convenient in certain scenarios, i.e., infections of the central nervous system [193], where only the free fraction of the drug would cross the blood-brain barrier, but it would also mean that a large amount of this drug would be rapidly excreted by the kidney. So that, after a prolonged period (24 h) the amount of free drug available would be the result of its constant release from binding to albumin, like when lower doses are administered. It should be noted, however, that our data came from measurements in healthy volunteers, and it is possible that in real patients with enterococcal IE, usually elderly and with a lower renal clearance rate, ceftriaxone levels would be higher during interdoses [194,195]. Unfortunately, our fears were confirmed by an unexpected failure rate (5/17; 29.4%) when we used the A + C24 regimen in the following patients. Although it is true that there are other factors that predispose to failure, such as the fact that the infection settles on prosthetic material (almost double the recurrence rate), or not performing surgery when there is a structural complication (e.g., an abscess of the ring), in our cohort we observed that patients who had complied with the A + C12 regimen while hospitalized for at least 3 weeks before switching to outpatient A + C24 had a much lower number of recurrences [196].

A recently reported pharmacokinetic/pharmacodynamic study simulating human doses of ampicillin and ceftriaxone has remarked the usefulness of this combination administering ceftriaxone every 12 h [197,198]. This prompted us to abandon this regimen and consider other alternatives such as the co-administration of ampicillin + ceftriaxone in the same infusion. We know that ampicillin is stable in infusion for more than 24 h. at room temperature, and there are abundant data reported on adequate plasma levels in both continuous and intermittent infusion [199,200,201]. However, there were doubts about the stability of ceftriaxone at room temperature and its interaction with other beta-lactams [202]. Using a very precise technique [203] we demonstrated the stability of this combination that we believe it would be an excellent alternative (much less toxic) to the A + G regimen, which is still used as a reference. Continuous or intermittent administration (every 4 h) of both drugs would always maintain levels above the required thresholds, without requiring any extraordinary manipulation other than periodic pump refilling [204,205] and have demonstrated excellent results in a few patients [196].

Other alternatives for OPAT in *E. faecalis* IE have been previously analyzed in a previous report [206].

## 6. Oral Treatment

Current guidelines for management of infective endocarditis (IE) usually recommend 4–6 weeks of IV antibiotics. This is based on historical data from animal models, which established the need for high peak serum antimicrobial levels that was thought to be only achievable with IV therapy. However, there has been an increasing recent interest in oral antibiotics as an alternative to prolonged parenteral therapy. Intravenous antibiotics offer high peak serum levels to be achieved rapidly. This is especially desirable in critically septic patients and is also a necessity in patients who are unable to take medications enterally, or when there are concerns about absorption. In addition, antimicrobial susceptibility may require antibiotics that can only be given IV, such as aminoglycosides or glycopeptides. However, antibiotics given orally with a good bioavailability and with favorable pharmacokinetic and pharmacodynamic properties with standard doses, would provide effective antimicrobial therapy for the treatment of endocarditis caused by susceptible microorganisms. The advantages compared with outpatient parenteral treatment is that oral antibiotic therapy may reduce costs and minimize challenges associated with OPAT including logistics, monitoring and risks of complications associated with intravenous catheters (e.g., bleeding, local ad systemic infections, and venous thrombosis. Reports of oral treatment in IE are scarce and mainly focused on streptococci [207,208,209] and staphylococci from right-sided IE [210,211,212,213,214,215,216]. More recently, a Danish trial [97,98] evaluated the efficacy and safety of sequential switching to oral antibiotic treatment in stable patients with IE after at least 10 days of parenteral therapy and, among patients who had undergone valve surgery for at least 7 days after the operation. From the 201 patients that were randomized to oral therapy, 51 (25.4%) were *E. faecalis* IE and the obtained primary outcome (a composite of all-cause mortality, unplanned cardiac surgery, embolic events, or relapses) seemed similar across the four different bacterial types (*E. faecalis*, streptococcus, *S. aureus* and coagulase-negative staphylococci). Regarding *E. faecalis*, four patients with oral treatment (4/51; 7.8%) presented the primary outcome compared with seven (7/46; 15.2%) in the parenteral treatment. To address the risk of subtherapeutical antibiotics levels related to potentially reduced gastrointestinal uptake, as well as the risk of variations in pharmacokinetics of the orally administered antibiotics, all oral regimens included two antibiotics from different drug classes, with different antibacterial effects, and different metabolization processes. Indeed, in seven patients in the orally treated group, the plasma concentration of one of the two administered antibiotics was not at the most effective level, as assessed by peak levels and time above the MIC, although the plasma concentration of the other simultaneously administered antibiotic was appropriate. The primary outcome did not occur in any of these patients and no antibiotic regimen were changed due to this finding.

However, despite its favorable results that have maintained over time [217,218], it should be remarked that only 400 (20%) of the 1954 screened patients underwent randomization, and that patients were orally treated a median of 17 days after intravenous treatment. Furthermore, regarding *E. faecalis* IE 15 patients (29.4%) in the oral group previously went to heart valve surgery. Other aspect is the combinations of high doses of amoxicillin, linezolid, rifampin, or moxifloxacin that may represent a challenge in elderly patients most prone to associated side effects. Thus, precaution is necessary when interpreting the POET trail regarding *E. faecalis* IE and further clinical trials focused on this group seems necessary for stablishing a robust recommendation.

## 7. Conclusions and Future Perspective

Although aminoglycoside containing regimens have been the standard of enterococcal IE treatment, the rise in resistance and availability of less nephrotoxic agents have led to novel treatment options. Double beta-lactam therapies have emerged as a novel strategy in the treatment of serious high-inoculum enterococcal infections due to their favorable side effect profiles and tolerability during long-term use. Currently, ampicillin plus ceftriaxone is the only beta-lactam therapy supported by robust clinical data and the main option for HLAR strains and elderly population, although other beta-lactam combinations supported by in vitro studies could be possible and must be explored. However, no recommendation can be done for non-HLAR strains that could also benefit from treatment with ampicillin plus a short course of gentamicin (2 weeks) and randomized trials will be necessary for a solid recommendation. Combinations of beta-lactams with daptomycin or fosfomycin are promising but needs further investigation. Other alternatives such as teicoplanin (with or without gentamycin), are interesting, especially in patients allergic to beta-lactams, because its lower renal toxicity and once-daily administration that favors OPAT. In this setting, the development of long-acting lipoglycopeptides (dalbavancin and oritavancin) has represented a considerable advance in security and shortening hospitalization times. Finally, oral treatment combining amoxicillin with linezolid or fluoroquinolones is also an alternative for continuation treatments but requires further investigation in this type of IE.

## Figures and Tables

**Table 1 antibiotics-12-00704-t001:** Antimicrobial resistance in enterococci.

Antimicrobial Class	Mechanism of Resistance Genes/Operons	Species	Comments
Beta-lactams -High level resistance * but susceptibility to beta-lactamase inhibitors -High level resistance **	Production of constitutive bet-lactamase or plasmid-mediated -Low-affinity PBP4 -Overproduction of PBP5	*E. faecalis* (rare in *E. faecium*) *E. faecalis* *E. faecium*	*: High level resistance to penicillin, ampicillin, and piperacillin. Sensitive to beta-lactamase inhibitors and carbapenems **: High level resistance to penicillin, ampicillin, piperacillin and carbapenems
Aminoglycosides -Low-level resistance -High-level resistance	-Defective aerobic transport across the cell membrane -Modification of the aminoglycoside by different enzymes (phosphotransferase, nucleotidyl transferase, acetyltransferase.) induced by self- transferable plasmids -Alteration of the ribosomal target site	*Enterococcus* spp.	-Structural resistance in all species -Resistance to several (gentamycin) or all aminoglycosides -Chromosomal mutation. Confers resistance to Streptomycin
*Glycopeptides (principal phenotypes)* -VanA (R to high levels of vancomycin and teicoplanin) -VanB (R to variable levels of vancomycin and susceptible to teicoplanin) -VanC (Low level R to vancomycin and susceptible to teicoplanin) -VanD (R to intermediate levels of vancomycin and low levels of teicoplanin) -VanE (R to low levels of vancomycin and susceptible to teicoplanin) -VanG (R to low levels of vancomycin and susceptible to teicoplanin) -VanN (R to low levels of vancomycin and susceptible to teicoplanin) -VanM (R to high levels of vancomycin and teicoplanin)	Transposons inserted into the chromosome or on plasmids. Induced by either vancomycin or teicoplanin Transposons inserted into the chromosome or on plasmids. Induced only by teicoplanin Located in the chromosome and non-transferable. Located in the chromosome and non-transferable. Expressed constitutively. Located in the chromosome and not transferable. Located on the chromosome and transferable. Expressed constitutively. Could be transferred by conjugation. Plasmid-encoded resistance. Could be transferred by conjugation	*E. faecium, E. faecalis* *E. faecium, E. faecalis* *E. gallinarum,* *E. casseliflavus, E. flavescens* *E. faecium* *E. faecalis* *E. faecium* *E. faecalis* *E. faecalis* *E. faecium* *E. faecium*	Synthesis of peptidoglycan precursors with low affinity for pentaglycopeptides ending in D-Ala-D-lac instead of D-Ala-D-Ala Synthesis of peptidoglycan precursors with low affinity for pentaglycopeptides ending in D-Ala-D-Ser instead of D-Ala-D-Ala
Quinolones	-Alterations in the GyrA or GyrB subunits of DNA gyrase (gene gyrA/gyrB) and/or the partC subunit of DNA topoisomerase Iv (gene parC) -qnr’-like gene -Efflux pump	*E. faecalis* *E. faecalis, E. faecium* *E. faecalis, E. faecium*	Most resistant strains have mutations in the two genes, that may be followed by additional mutations that increase the level of resistance. -Encodes protein that protects DNA gyrase from inhibition by the drug -Efflux pumps are yet not identified but are strongly suspected (Low-level resistance)
Oxazolidines	-rRNA genes *-cfr* (plasmid-mediated) *-optr*A (plasmid-mediated)	*E. faecalis, E. faecium* *E. faecalis, E. faecium* *E. faecalis, E. faecium*	-Multiple mutations reducing affinity in ribosomal subunit. -Methylation of 23S rRNA. Confers resistance to other antimicrobials. -Encodes an ABC transporter
Daptomycin	-YycFGHIJ/LiaFSR -Mutation in gene encoding cardiolipin synthase	*E. faecalis, E. faecium*	-Stress-sensing response system -Alteration in membrane charge and fluidity
Macrolides and clindamycin	Methylation of an adenosine residue in the 23S rRNA (gene *erm*B)	*Enterococcus* spp.	Located on plasmids and transposons. Constitutive or inducible. Confers resistance to various antimicrobials
Tetracyclines	-Active eflflux (gene *tet*L) -Protection of the ribosome (genes *tet*N and *tet*M) -Overexpression of gene *tet*(L) and *tet* (M)	*Enterococcus* spp. *Enterococcus* spp. *E. faecium*	-Low-level resistance -Located on plasmid and transposons. Constitutive or inducible. Confers high-level resistance. -Reported with Tigecycline
Chloramphenicol	Chloramphenicol acetyltransferase (gene *cat*)	*Enterococcus* spp.	Produces acetylation of chloramphenicol. Plasmid-mediated
Sulphonamides and trimethoprim	No gene identified.	*Enterococcus* spp.	Enterococci can use exogenous folates

**Table 2 antibiotics-12-00704-t002:** American Heart Association [65] and European Society of Cardiology [66] guidelines for the treatment of Enterococcus infective endocarditis.

Indication	Recommendation	Dosage and Route	Duration (Weeks)	Comments
Strains susceptible to Penicillin and Gentamicin in patients who can tolerate β-Lactam therapy	Ampicillin/amoxicillin or penicillin G plus gentamicin	2 g IV every 4 h 18–30 million U/24 h IV either continuously or in 6 equally divided doses 3 mg/kg ideal body weight in 2–3 equally divided doses IV/IM [65] or in one dose [66]	4–6 4–6 2–6	4-wk therapy recommended for patients with native valve and symptoms of illness < 3 months. 6-wk therapy recommended for native valve symptoms > 3 months and for patients with prosthetic valve or prosthetic material. - Gentamycin is less toxic in single administration eviting drug levels monitorization and could be given for two weeks with similar efficacy [66].
	Ampicillin plus ceftriaxone	2 g IV every 4 h 2 g IV/IM every 12 h	6 6	- This combination is active against *E. faecalis* strains with and without HLAR *, being the combination of choice in patients with HLAR *E. faecalis* endocarditis. - Recommended for patients with initial creatinine clearance < 50 mL/min or who develop creatinine clearance < 50 mL/min during therapy with gentamicin-containing regimen. This combination is not active against *E. faecium*
Strains susceptible to Penicillin and resistant to Aminoglycosides or Streptomycin-susceptible/ Gentamicin-resistant in patients able to tolerate β-Lactam therapy [65]	Ampicillin plus ceftriaxone	2 g IV every 4 h 2 g IV/IM every 12 h	6 6	
	Ampicillin or penicillin G plus Streptomycin ^#^	2 g IV every 4 h 18–30 million U/24 h IV either continuously or in 6 equally divided doses 15 mg/kg ideal body weight per 24 h IV or IM in 2 equally divided doses	4–6 4–6 4–6	Use is reasonable only for patients with availability of rapid Streptomycin serum concentrations. Patients with creatinine clearance < 50 mL/min or who develop creatinine clearance < 50 mL/min during treatment should be treated with double–β-lactam regimen. Patients with abnormal cranial nerve VIII function should be treated with double–β-lactam regimen.
Patients unable to tolerate β-Lactam or Penicillin-resistant *Enterococcus* species and Aminoglycoside-susceptible strains	Vancomycin plus gentamycin ^¶^	30 mg/kg per 24 h IV in 2 equally divided doses 3 mg/kg per 24 h IV in 3 equally divided doses [65] or in a single dose [66]	6 6	For β-lactamase–producing strain, if able to tolerate a β-lactam antibiotic, ampicillin-sulbactam plus aminoglycoside therapy may be used ^§^ If resistance is due to PBP5 alteration, use vancomycin-based regimens
*Enterococcus* species caused by strains resistant to Penicillin, Aminoglycosides, and Vancomycin	Linezolid or Daptomycin (plus) Ampicillin	600 mg IV or orally every 12 h 10–12 mg/kg IV per dose 200 mg/kg/day IV in 4–6 doses	>6 >6	Linezolid use may be associated with potentially severe bone marrow suppression, neuropathy, and numerous drug interactions. haematological toxicity must be monitored. Patients with IE caused by these strains should be treated by a care team including specialists in infectious diseases, cardiology, cardiac surgery, clinical pharmacy, and, in children, pediatrics. Cardiac valve replacement may be necessary for cure.
	Quinupristin–dalfopristin	7.5 mg/kg IV every 8 h	>6	Quinupristin–dalfopristin is not active against *E. faecalis*

Doses are recommended for patients with normal renal and hepatic function. Paediatric doses: ampicillin 300 mg/kg/day, ceftriaxone 100 g/kg/12 h IV or IM, vancomycin 40 mg/kg/day IV in 2–3 equally divided doses *: HLAR: High-level resistance to aminoglycosides. #: High-level resistance to gentamicin (MIC > 1024 mg/L): if susceptible to streptomycin (MIC < 500 mg/L), replace gentamicin with streptomycin 15 mg/kg/day in two equally divided doses. Streptomycin dose should be adjusted to obtain a serum peak concentration of 20 to 35 µg/mL and a trough concentration of <10 µg/mL. ¶: Dose of gentamicin should be adjusted to obtain serum peak and trough concentrations of 3 to 4 and <1 µg/mL, respectively. Dose of vancomycin should be adjusted to obtain a serum trough concentration of 10 to 20 µg/mL. §: Ampicillin-sulbactam dosing is 3 g/6 h IV.

**Table 3 antibiotics-12-00704-t003:** Alternatives for treatment in *Enterococcus* infective endocarditis.

Recommendation	Dosage and Route	Duration (Weeks)	References	Comments
**Beta-lactam combined**				
Ampicillin plus Ciprofloxacin/ofloxacin	2 g IV every 4 h 500 mg/8–12 h	6–8	[67,68]	Very few experience in *E. faecalis* bacteremia
Ceftobiprole or Ceftarolin With/without gentamycin	500–1000 mg/12 h IV 400 mg IV every 12 h 3 mg/kg/day IV or IM	6–9 2	[28,69,70]	Only in vitro studies and in experimental model of peritonitis
**Lipo/Glucopeptides**				
Teicoplanin with/without gentamycin	6–10 mg/Kg/d IV or IM per dose 3 mg/kg/day IV or IM	6–8 2	[71,72,73,74,75]	Good results in experimental models and in humans as sequential therapy for IE
Dalbavancin	1500 mg in a single dose IV, then 1000 mg every two weeks	6–8	[76,77,78]	Good results in humans as sequentially therapy for IE
Oritavancin	1200 mg in a single dose IV, then 800 mg every week	6–8	[79,80,81]	Very scarce experience reported in human IE
**Daptomycin-based regimens**				
Daptomycin plus ceftaroline or ceftobiprole	10–12 mg/kg IV per dose 400 mg IV every 12 h 500 mg IV every 8 h	6–8	[69,70,82,83,84]	Synergistic effects in vitro and experimental models. Very few experience in human IE.
Daptomycin plus imipenem	10–12 mg/kg IV per dose 1 gr IV every 6 h	6–8	[84]	Synergistic effects in vitro and experimental models
Daptomycin plus Fosfomycin with/without gentamycin	10–12 mg/kg IV per dose 3 g IV every 6 h 3 mg/kg/day IV or IM	6–8	[85,86,87,88,89,90,91]	Synergistic effects in vitro and experimental models. Good results for *S. aureus* IE
Daptomycin plus Tigecycline	10–12 mg/kg IV per dose 50–100 mg/12 h	6–8	[92,93,94]	Always Consider the use of higher doses of tigecycline
**Oxazolidines**				
Tedizolid	200 mg IV or PO/24 h		[67,68,95,96]	Less toxic and more active in vitro than linezolid. There is no experience among IE in humans
**Oral regimens**				
Amoxicillin plus moxifloxacin	1 g. every 6 h 400 mg every 12 h	4–6	[97,98]	Oral treatment must be considered as a sequentially strategy in selected patients.
Amoxicillin plus Linezolid	1 g. every 6 h 600 mg very 12 h	4–6	[97,98]	Oral treatment must be considered as a sequentially strategy in selected patients.
Amoxicillin plus Rifampicin	1 g. every 6 h 600 mg every 12 h	4–6	[97,98]	Oral treatment must be considered as a sequentially strategy in selected patients.
Moxifloxacin plus Linezolid	400 mg every 12 h 600 mg very 12 h	4–6	[97,98]	Oral treatment must be considered as a sequentially strategy in selected patients.

**Table 4 antibiotics-12-00704-t004:** Adjustment of antimicrobials according to the Glomerular Filtration Rate (GFR). Doses are provided for the treatment of *E. faecalis* endocarditis.

Antimicrobial	GFR > 50	GFR = 50–30	GFR = 30–10	GFR < 10 ^¶^
**Beta-lactams**				
Penicillin	4 M UI/4 h IV	4 M UI/6 h IV	3 M UI/8 h IV	2 M UI/12 h IV
Ampicillin	2 g/4 h IV	2 g/6 h IV	2 g/8 h IV	1 g/12 h IV
Amoxicilin	2 g/4 h IV 1 g/6 h po	2 g/6 h IV 1 g/8 h po	2 g/8 h IV 0.5–1 g/12 h po	1 g/12 h IV 0.5–1 g/24 h po
Ceftriaxone	2 g/12 h IV	2 g/12 h IV	2 g/12 h IV	2 g/24 h IV
Ceftriboprole	500–1000 mg/12 h IV	500 mg/12 h IV	250 mg/12 h IV	250 mg/12 h IV
Ceftaroline	400 mg/8–12 h IV	400 mg/8–12 h IV	300 mg/8–12 h IV	200 mg/8–12 h IV
Imipenem	1 g/6–8 h IV	0.5–1 g/8 h IV	0.5 g/8–12 h IV	0.5 g/12 h IV
Meropenem	1 g/6–8 h IV	1 g/8 h IV	1 g/12 h IV	1 g/24 h IV
**Aminoglycosides**				
Gentamycin *	3 mg/kg/24 h IV	3 mg/kg/24–36 h IV	3 mg/kg/48 h IV	1–2 mg/kg/48 h IV
Glucopeptides				
Vancomycin *	15–20 mg/kg/8–12 h IV	15 mg/kg/24 h IV	15 mg/kg/24–48 h IV	15 mg/kg/72 h IV
Teicoplanin *	10–12 mg/Kg/24 h IV	10–12 mg/kg/48 h IV	10–12 mg/kg/72 h IV	10–12 mg/kg/72 h IV
**Gluco/lipo peptides**				
Daptomycin	8–12 mg/Kg/24 h IV	8–12 mg/kg/24 h IV	8–12 mg/kg/48 h IV	8–12 mg/kg/48 h IV
Dalbabancin	1500 mg/2 weeks IV	1000–1500 mg/2 weeks IV	1000 mg/2 weeks IV	1000 mg/2 weeks IV
Oritavancin	1200 mg/2 weeks IV	1200 mg/2 weeks IV	1200 mg/2 weeks IV	1200 mg/2 weeks IV
**Oxazolidines**				
Linezolid	600 mg/12 h IV/po	600 mg/12 h IV/po	600 mg/12 h IV/po	600 mg/12 h IV/po
Tedizolid	200 mg/24 h IV/po	200 mg/24 h IV/po	200 mg/24 h IV/po	200 mg/24 h IV/po
**Quinolones**				
Ciprofloxacin	400 mg/8 h IV 750 mg/12 h po	400 mg/12 h IV 500 mg/12 h po	200 mg/8–12 h IV 500 mg/12 h po	400 mg/24 h IV 750 mg/24 h po
Levofloxacin	500 mg/12 h IV/po	500 mg/24 h IV/po	250 mg/24 h IV/po	250 mg/24–48 h IV/po
Moxifloxacin	400 mg/24 h po	400 mg/24 h po	400 mg/24 h po	400 mg/24 h po
**Others**				
Fosfomycin	4 g/8 h IV	4 g/12 h IV	4 g/24 h IV	2–4 g/24–48 h IV
Tigecycline	50–100 mg/12 h IV	50–100 mg/12 h IV	50–100 mg/12 h IV	50–100 mg/12 h IV
Rifampin	600–1200 mg/24 h IV/po	600–1200 mg/24 h IV/po	600 mg/24 h IV/po	600 mg/24 h IV/po

*: Drug levels must be monitored. ¶: In patients on Haemodialysis (HD) doses IV must be administered after the HD session, excepting gentamycin that must be administered at the beginning, monitoring peak and valley levels.

## Data Availability

Not applicable.

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
