# Peer review of "Treatment of *Enterococcus faecalis* Infective Endocarditis: A Continuing Challenge"

_antibiotics, 2023, doi:10.3390/antibiotics12040704_

Round 1

Reviewer 1 Report

General comments

This manuscript is a review on enterococcal endocarditis. Mechanisms of resistance are described for the relevant drugs as is outpatient antibiotic therapy (OPAT) and duration of treatment. Tables show antimicrobial resistance, AHA- and ESC-recommendations for the management of enterococcal endocarditis and alternative treatments published in in vitro and in vivo studies.

Enterococcal endocarditis has been a problem for decades. New drugs have emerged and new combinations have been used. However, since the introduction of the combination of ampicillin with ceftriaxone for the treatment of infectious endocarditis due to E. faecalis no major steps forward have been made. The authors have considerable experience with the latter combination and its discussion in the OPAT-section is one of the strengths of this review. I acknowledge that a review like this on enterococcal endocarditis has not been written during the last years. However, it should be made clearer why this review is important now. Do we have new drugs or new findings or a new situation? We have guidelines for the treatment of endocarditis. They are from 2015. Where do they need to be reconsidered? What exactly is the knowledge gap? In addition, the structure of the manuscript needs improvement. There are two chapters: 1. introduction, 2. mechanisms of resistance. However, therapeutic choices (2.8.), duration of treatment (2.18.), OPAT (2.19.) come under the heading "mechanisms of resistance". This needs to be adapted. The tables contain many details but need adaptation. Table 1 is not well readable, tables 2 and 3 should be combined in a single table. Table 4 is well readable but should be grouped (see details below).

Specific comments

Table 1: not well readable

I suggest columns for antimicrobial class, mechanism of resistance, species, clinical relevance (much information from the first column could be transferred to this column), and comments (the comments now include many issues related to resistance. This information should preferably provided in the resistance column).

Line 143-144: the evidence that teicoplanin is not associated with renal toxicity is at least conflicting. You could omit this part of the phrase or attenuate this statement and provide references. I believe that therapeutic drug monitoring with teicoplanin is essential as it is with vancomycin and that nephrotoxicity is dependent on exposure.

Table 2 + table 3

The information is copied (with appropriate references) from the 2015 AHA- and ESC-guidelines for the management of infective endocarditis. I believe that copying the reccommendations is of little value. You could provide a synthesis of the two guidelines in a single table (there is not much substantial difference) and highlight the differences and later give your own recommendation (as you did for the once-daily use of aminoglycosids and the duration of administration of aminoglycosids).

Table 4

Fine and interesting work. I suggest to group the various regimens (i.e. daptomycin-based, ß-lactam-based, oxazolidinone-based, others).

Line 259-267: you are right that the combination of penicillins and aminoglycosids have been considered standard of care for decades. As this is a review on enterococcal endocarditis I suggest to highlight the lack of data. To the best of my knowledge there has for instance never been an RCT confirming the superiority of this regimen. This a clear knowledge gap.

Line 325: it should be "Clostridioides" instead of "Clostridium".

Line 351: see above. Please provide data for the lack of renal toxicity of teicoplanin or attenuate this statement. I agree that once daily dosing may be a clear benefit when outpatient treatment is considered.

Line 558: the two-week treatment with aminoglycosids leads to less nephrotoxicity. I believe this was an inadvertent mistake.

Reviewer 2 Report

It is necessary to look more closely at groups of patients who are at increased risk of developing infective endocarditis, as well as the behavior of antibiotic doses according to the degree of renal failure, considering that the hemodialysis population belongs to high-risk groups.too many references reporting identical results

Reviewer 3 Report

This manuscript is a very complete review of the management of antimicrobial therapy in patients with enterococcus faecalis infective endocarditis including many references. However, in my opinion, the main drawback of this manuscript is its length. The chapter on resistance mechanisms is long and too detailed. It could be halved without affecting the content of the text. Most particularly, all comments about E. Faecium are off topic (  exemple : page  4 line 112-120)

Minor comments :

- Just after table 4 : 2.9 beta with-lactam aminoglycosides

- Page 2 line 62- 63 Comments concerning E Faecium can be suppressed

- Page 9 line 296 :  2 g IV/12 hrs

- Page 10 line 345-350 : this comment concerns combination between penicillin and gentamicin and production of inducible beta-lactamase. This is not related with glycopeptides and must be suppressed

- Page 12 line 430  endocarditis and not enfocarditis

- Page 13 line 506  OPAT is not defined previously in the manuscript

- Page 13 line  504- 513. This comment is unclear. Two studies are reported with 27 and 34 patients respectively. Only six patients were successfully treated and in the next line authors suggest that dalbavancin might be a good option to treat E Faecalis IE. This results seems weak for such a recommendation.

- Page 14 line 558 :  Concerning the study referenced 166. The authors report that there is a reduction in renal function with the two veeks regimen of gentamicin. Is not rather the opposite with a longer duration of gentamicin treatment associated with a worse outcome ?

- Page 16 line 639. « This prompted usto abandon this regimen and consider other alternatives ». Can you specify which regimen and which alternatives ? this is unclear

Round 2

Reviewer 1 Report

Thank you for your revision. You have appropriately addressed most of the issues I had raised in my review. The tables in particular are more concise.

Author Response

Thank you again, sir.

Reviewer 2 Report

The decision number of the Ethics Committee is missing!

Author Response

This a review. No decision of the Ethics Comittee is required